# Leveraging Multiple Teachers for Test-Time Adaptation of Language-Guided Classifiers

**Kangda Wei**[1]     **Sayan Ghosh**[2]     **Rakesh R. Menon**[2]     **Shashank Srivastava**[2]

[1]Department of Computer Science and Engineering, Texas A&M University

[2]Department of Computer Science, UNC Chapel Hill

kangda@tamu.edu   {sayghosh, rrmenon, ssrivastava}@cs.unc.edu

## Abstract

Recent approaches have explored language-guided classifiers capable of classifying examples from novel tasks when provided with task-specific natural language explanations, instructions or prompts (Sanh et al., 2022; R. Menon et al., 2022). While these classifiers can generalize in zero-shot settings, their task performance often varies substantially between different language explanations in unpredictable ways (Lu et al., 2022; Gonen et al., 2022). Also, current approaches fail to leverage unlabeled examples that may be available in many scenarios. Here, we introduce TALC, a framework that uses data programming to adapt a language-guided classifier for a new task during inference when provided with explanations from multiple teachers and unlabeled test examples. Our results show that TALC consistently outperforms a competitive baseline from prior work by an impressive 9.3% (relative improvement). Further, we demonstrate the robustness of TALC to variations in the quality and quantity of provided explanations, highlighting its potential in scenarios where learning from multiple teachers or a crowd is involved. Our code is available at: https://github.com/WeiKangda/TALC.git.

## 1 Introduction

Inductive learning from examples has underpinned many successful machine learning applications. However, classifiers trained solely from labeled examples often struggle to generalize in scenarios with limited labeled data. In contrast, humans can learn new concepts through natural language conversations (Chopra et al., 2019; Tomasello, 1999). Inspired by this phenomenon, recent approaches use natural language explanations, instructions, and prompts to train *language-guided classifiers* (Srivastava et al., 2017; Andreas et al., 2018; Murty et al., 2020; Wang* et al., 2020; Ye et al., 2020). While these classifiers can perform zero-shot classification, they have several limitations. Firstly, they

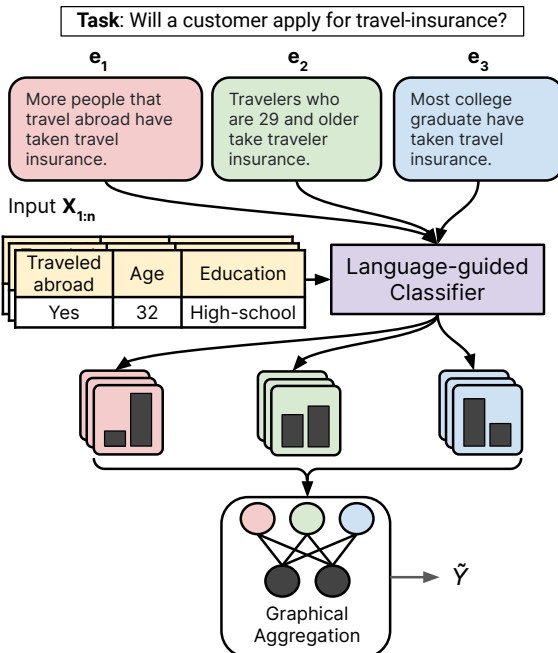

Figure 1: TALC leverages data programming to perform test-time adaptation of a language-guided classifier. Natural language explanations ($E = \{e_1, e_2, e_3\}$) provided by multiple teachers and unlabeled examples ($X_{1:n}$) for a new task are fed to a language-guided classifier pair-wise resulting in multiple pseudo-labels for the unlabeled examples. TALC uses a graphical aggregation to weigh the pseudo-labels from different explanations to decide the final predicted label ($\hat{Y}$). TALC is highly flexible in aggregating labels as it can conceptually consider a broad variety of factors, such as the complexity of explanations, consistency between explanation predictions, identity of the explanation provider, etc.

lack a principled strategy for weighing language supervision from multiple sources (or teachers). Secondly, they fail to utilize the available unlabeled data for a new task during inference. Additionally, the impact of the quality of sources, and the inclusion of low-quality explanations, remains largely unexplored.

To address these limitations, we present TALC (**T**est-time **A**daptation of **L**anguage-guided **C**lassifiers), a framework for adapting language-

guided classifier on novel tasks during inference, also known as test-time adaptation. TALC assumes a priori access to the entire test set (unlabeled samples) of the novel task and the task-specific explanations, which aligns with real-world situations, such as developing a product-category classifier for an e-commerce platform. In the context of TALC, the multiple explanations available for each task are considered as distinct supervisory signals. Leveraging the power of data programming (Ratner et al., 2018b), TALC aggregates and combines the supervision provided by these multiple explanations effectively.

Figure 1 illustrates the TALC framework. TALC uses a subset of the test data, called the adaptation set, for adapting the language-guided classifier. For each pair of explanation and test example in the adaptation set, a pseudo-label is generated using the base language-guided classifier. TALC learns a label aggregator on the pseudo-labels generated for the adaptation set using EM (§3). The label aggregator is trained to consider the contribution of each explanation during adaptation, thus, in principle, allowing it to weigh different sources of language supervision. Finally, TALC uses the learned aggregator over the entire test set to obtain final predictions for test set examples.

We evaluate TALC on six classification tasks from the CLUES-Real dataset (R. Menon et al., 2022), where each task is paired with natural language explanations (§4). TALC outperforms strong baselines by 3.3% on average (absolute). Through qualitative and quantitative analysis, we investigate TALC's robustness with respect to the size of the adaptation set, number of explanations, and explanation quality. In the subsequent sections, we describe TALC in detail (§3), present experimental results and analysis (§4), and conclude by discussing our contributions, limitations, and ethical considerations. Our contributions are:

- We introduce TALC, a test-time adaptation framework, that uses label aggregation to improve language-guided classifiers.
- We demonstrate the effectiveness of TALC on multiple real-world classification tasks from CLUES-Real (R. Menon et al., 2022).
- We present comprehensive analyses to evaluate the robustness of TALC w.r.t. the quantity and quality of explanations.

## 2 Related Work

**Learning From Language**    Using natural language explanations to inform or train classifiers has garnered significant interest in recent years (Goldwasser and Roth, 2014; Srivastava et al., 2017; Hancock et al., 2018; Murty et al., 2020). While Murty et al. (2020) enhance supervised BERT models (Devlin et al., 2019) for relation extraction tasks, other approaches employ language explanations for few-shot learning. For instance, Hancock et al. (2018) convert explanations to labeling functions via semantic parsing, leveraging unlabeled data for weak labels. More recently, R. Menon et al. (2022) utilize natural language explanations in an entailment-based model for classification decisions.

**Test-time Adaptation**    Test-time adaptation has been extensively studied in computer vision by employing batch-normalization statistics (Nado et al., 2020; Khurana et al., 2021; Schneider et al., 2020), test-time entropy minimization (Wang et al., 2020; Sivaprasad and Fleuret, 2021), prediction consistency maximization over augmentations (Zhang et al., 2021), and classifier adjustment (Iwasawa and Matsuo, 2021). In the realm of natural language processing, Banerjee et al. (2021) explore test-time adaptation for question-answering using self-supervision. In contrast, we introduce a new test-time adaptation approach that leverages data programming to adapt a base language-guided classifier during inference by utilizing natural language explanations.

**Data Programming**    Data programming (Ratner et al., 2017) employs a combination of multiple labeling functions and generative models to create probabilistic training labels for unlabeled datasets. Prior work (Ratner et al., 2018b; Hancock et al., 2018) has demonstrated successful applications of this paradigm to create systems that allow users to label large datasets programmatically. Here, we repurpose data programming in the test-time adaptation setting to improve classifiers on unseen tasks.

## 3 TALC

In this section, we present the details of our framework, TALC. TALC leverages data programming to adapt a base natural language explanation-guided classifier on a novel task during inference.

**Problem Setup.**    We assume a language-guided classifier, $\mathcal{M}_{LC}$, which can take an explanation $e$

from a teacher and example $X$ to predict a label $\mathcal{M}_{LC}(X, e)$. A language-guided classifier refers to a classifier that utilizes one or more natural language explanations to make predictions. Our objective is to make predictions for a batch of test samples, represented as $\{X_{test}, Y_{test}\}_{1:n}$, where $Y_{test}$ represents the unobserved ground-truth labels corresponding to the samples in $X_{test}$, and $n$ denotes the number of samples. During test-time adaptation, our aim is to effectively adapt the classifier to the specific task at hand and infer the true labels for $X_{test}$. Existing methods for test-time adaptation typically assume an online setting, where examples are processed one at a time (Sun et al., 2020; Banerjee et al., 2021). In contrast, we assume a priori access to the entire test set of the task. This assumption allows us to leverage the empirical distribution of the unlabeled data for semi-supervised learning.

Our setting aligns with real-world scenarios, such as developing a product-category classifier for an e-commerce platform, where the complete database of products (including the test set) is known in advance. For situations where test samples are observed one at a time, it is still possible to utilize TALC for adapting a base classifier. This involves a "warm-up" phase, where the base classifier is used off-the-shelf for a few samples, followed by adaptation using TALC. While this usage scenario is not the primary focus of our work, we provide a description of how TALC can be employed in such cases in Appendix §A for brevity.

**Overview.** As depicted in Figure 1, for a new task $T_{new}$, we are provided with $m$ natural language explanations $E = \{e_1, e_2, \dots, e_m\}$, and a set of examples $\{X_i \in X_{test}\}$. To generate the classifier outputs, we iterate through each explanation $e_j$ for every example $X_i$, and compute $M_{ij} := \mathcal{M}_{LC}(X_i, e_j)$. This yields a labeling matrix $M$ with a shape of $n \times m$. Next, we introduce a test-time adaptation procedure: TALC, to compute the final labels $\tilde{Y}$ utilizing the $M$. This procedure essentially implements a function $f : M \in \mathbb{R}^{n \times m} \to \tilde{Y} \in \mathbb{R}^n$, which we describe in the rest of this section.

**Test-time Adaptation.** The objective of TALC is to adapt the language-guided classifier, $\mathcal{M}_{LC}$, on a novel task, $T_{new}$ during inference. We illustrate the adaptation procedure in Algorithm 1. First, we split the test set into two disjoint sets - the adaptation

---

**Algorithm 1 TALC**

   **Inputs:** Language-guided classifier $\mathcal{M}_{LC}$, test set $X_{test}$, task explanations $E$, adaptation ratio $\alpha$
1: $N_{adapt} \leftarrow \alpha \times |X_{test}|$
2: $X_{test}^{adapt} \leftarrow X_{test}[: N_{adapt}]$
3: $X_{test}^{held-out} \leftarrow X_{test}[N_{adapt} :]$
4: Train the label aggregator $\mathcal{L}_w^{\mathrm{agg}}$ on $X_{test}^{adapt}$
   $\hat{w} \leftarrow \underset{w}{\arg\max}\, P_w(X, E; \mathcal{M}_{LC}$ (using EM)
5: Infer $\tilde{Y}_{TALC}$ for $X_{test}$ using the learned $\hat{w}$.
   $\tilde{Y}_{TALC} := \underset{Y}{\arg\max}\, P_{\hat{w}}(Y|X_{test}, E, \mathcal{M}_{LC})$

   **return** $\tilde{Y}_{TALC}$

---

set and the held-out test. The adaptation set is utilized by TALC to adapt $\mathcal{M}_{LC}$. The proportion of the test set that forms the adaptation set is defined by an adaptation ratio, $\alpha \in [0, 1]$, defined as $\alpha = \frac{|\text{adaptation set}|}{|\text{test set}|}$ We also partition the labeling matrix $M$ into $M^{adapt}$ and $M^{held-out}$ by choosing the rows corresponding to samples in the adaptation set and held-out set, respectively.

To model the dependence between the (latent) inferred labels and $M^{adapt}$, we use data programming techniques (Ratner et al., 2019) to train a label aggregator, $\mathcal{L}_w^{\mathrm{agg}}$, with task-specific parameters $w$. We use the learned parameters (which correspond to weights learned for each explanation) to aggregate predictions in $M$ (both $M^{adapt}$ and $M^{held-out}$) and infer the labels, $\tilde{Y}_{TALC}$.

**Label Aggregator.** The label aggregator is a graphical model that defines the joint probability of explanations $E$, examples $X$ and latent (true) labels $Y$ for a given language-guided classifier as:

$$P(X, E, Y; \mathcal{M}_{LC}) \propto \exp w^T \phi(X, E, Y, \mathcal{M}_{LC}) \tag{1}$$

Here, $\phi$ is a feature-vector of features that can be computed using $X, E, Y$ and $\mathcal{M}_{LC}$ and $w$ is a weight vector corresponding to each of those features. In general, this can subsume a very broad range of features, including features that can indicate the complexity of an explanation, or its provenance [1]. We also note that in particular, since the labeling matrix $M$ is computed from $X$, $E$ and $\mathcal{M}_{LC}$, $\phi$ can include features that depend on $M$ and $Y$. For simplicity, our instantiation incorporates the labeling rates of each explanation (how

---

[1] So, for example, the aggregator can automatically learn to lean more/less on complex explanations, or trust explanations from specific sources more than from others

frequently an explanation assigns a label[2]) and the correlations between the pseudo-labels from different explanations to estimate the accuracies of each individual explanation in an unsupervised manner. Specifically, the label aggregator is defined in terms of two types of features: accuracy ($\phi^{Acc} \in \mathbb{R}^{n \times m}$) and propensity ($\phi^{Prop} \in \mathbb{R}^{n \times m}$). Each value in $\phi^{Acc}$ and $\phi^{Prop}$ is defined as:

$$\phi_{i,j}^{Acc}(M, Y) = \mathbb{1}\{M_{i,j} = y_i\} \qquad (2)$$

$$\phi_{i,j}^{Prop}(M, Y) = \mathbb{1}\{M_{i,j} \neq y_{abstain}\} \qquad (3)$$

where $y_i$ is the label for $i^{th}$ sample and $y_{abstain}$ is a special label that denotes $\mathcal{M}_{LC}$ has abstained from predicting a label based on the $j^{th}$ explanation. The accuracy factor fires if the inferred label ($Y_i$) for an unlabeled example $X_i$ matches the predicted label from an explanation $j$. The propensity factor fires whenever the classifier doesn't abstain from predicting a label from an explanation.

Since here we only define two types of features for each explanation, $w \in \mathbb{R}^{2m}$ is a learnable vector corresponding to weights for the accuracy factor and propensity factor for each explanation. The weights are learned by maximizing the log-likelihood $\log P(X, E) = \log \sum_Y P_w(X, E, Y)$ using the expectation-maximization (EM) algorithm (since we don't have ground-truth labels for $Y$ at test-time). We compute the MAP estimate $\tilde{Y}_{TALC} := \underset{Y}{\arg\max} P_{\hat{w}}(Y|X, E)$ using Gibbs sampling to predict the final labels. Note that while we learn the weights $\hat{w}$ on the adaptation set (line 4 in Algorithm 1), the learned weights are used to aggregate predictions in both the adaptation and the held-out examples, to predict the labels, $\tilde{Y}_{TALC}$ (line 5 in Algorithm 1). We implement the label aggregator using Snorkel-Metal[3] (Ratner et al., 2018a). Appendix §C provides task-specific details of the label aggregator training.

## 4 Experiment and Analysis

In this section, we evaluate the zero-shot adaptation performance of TALC on classification tasks, followed by a detailed analysis of TALC's robustness.

### 4.1 Data

We assess the performance of TALC on real-world classification tasks from the CLUES (R. Menon et al., 2022) benchmark. Out of the sixteen real-world tasks in the test split of CLUES, we focus on six tasks for evaluation due to the limited number of test samples ($< 10$) in the remaining tasks, which restricts their suitability for test-time adaptation. Figure 1 presents an illustrative example showcasing the nature of these tasks and provides examples of the corresponding natural language explanations. Appendix §B provides further details regarding the six tasks selected for evaluation.

We utilize the ExEnt model (R. Menon et al., 2022) as the base language-guided classifier ($\mathcal{M}_{LC}$) in alignment with our choice of the CLUES dataset[4]. The ExEnt model leverages textual entailment to establish the correspondence between explanations and tabular inputs, enabling label predictions. To aggregate predictions from multiple explanations, ExEnt adopts a mean-pooling strategy, aggregating the predictions obtained from each explanation-input pair to derive the final label. It's important to note that ExEnt is not trained for abstention, meaning it always assigns a label regardless of the quality of the explanations. In §4.4, we further explore the scenario of abstention, which we consider a more realistic use case for language-guided classifiers.

### 4.2 Baseline and Evaluation Metrics

We compare TALC against the following baselines:
1. ExEnt: This refers to the base ExEnt model (R. Menon et al., 2022) that has been trained on real-world training tasks from the CLUES dataset.
2. ExEnt-MV: For each example $X_i$, we generate a set of pseudo-labels corresponding to each of the $m$ task explanations. The final predicted label is determined by selecting the label that appears most frequently among the $m$ pseudo-labels (*majority vote*). Unlike ExEnt, which uses mean-pooling for aggregation, ExEnt-MV applies a mode-pooling operation.
3. ExEnt-FT: Similar to our approach of fine-tuning TALC with the predicted labels from the label aggregator, $\mathcal{L}_w^{\text{agg}}$, we also include a self-training baseline approach by *fine-tuning* ExEnt. This involves utilizing ExEnt's own predictions as labels on the adaptation set.

We use classification accuracy as the evaluation metric to compare the utility of different methods.

---

[2]The pseudo-label corresponding to an explanation can either be a class label, or a special label, $y_{abstain}$ (e.g. if an explanation does not apply for an example)

[3]https://www.snorkel.org/

[4]At the time of writing, this is the best model on CLUES with publicly available code.

## 4.3 Results

Table 1 shows the zero-shot classification accuracy of `TALC` and the baselines on the six evaluation tasks. The findings reveal several key insights. Firstly, we observe that majority voting (`ExEnt-MV`) performs better than vanilla `ExEnt` on average across the six tasks. Secondly, fine-tuning `ExEnt` on its own predictions (`ExEnt-FT`) results in better zero-shot accuracies than the base `ExEnt` model, demonstrating the value of self-training on unlabeled data. Furthermore, the performance of `ExEnt-FT` increases with an increase in the amount of test data used for adaptation ($35.7 \rightarrow 36.8$ as we increase the adaptation ratio from $0.5 \rightarrow 1.0$).

We note that `TALC` obtains better performance on average across all evaluation tasks compared to the three baselines. Specifically, `TALC` improves the accuracy by around $3.3\%$ on average (absolute) over the state-of-the-art `ExEnt` model. In fact, both `TALC` variants, at adaptation ratio 0.5 and 1.0, perform better than `ExEnt` on all tasks except for indian-liver-patient. The utilization of the label aggregator in `TALC` results the biggest improvement ($\sim 25\%$ relative) on the tic-tac-toe-endgame task. We attribute this improvement to the label aggregator's ability to give higher weightage to high-quality explanations, resulting in more accurate predictions. For the tasks where the performance of `TALC` is close for $\alpha = 0.5$ and $\alpha = 1.0$ in Table 1, we observed that the aggregation weights for each explanation learned by the data programming framework are roughly similar for the two settings. As a result, the aggregation over the pseudo labels in the labeling matrix produces similar final predictions and hence similar accuracies.

## 4.4 Analysis

**Abstention.** Our previous experimental results treat labels from individual explanations the same, irrespective of the confidence of the model in its predictions on those examples. This is because the base-language classifier used in experiments, `ExEnt`, always chooses a label during inference rather than performing selective predictions. However, the `TALC` framework allows for differential modeling of abstentions, where a model can choose to refrain from assigning a class label if the explanation does not apply to the example. To explore this, we design a variant of `ExEnt`, referred to as `ExEnt-A`, that can refrain from assigning a class label during inference. This is straightforward since

`ExEnt` is based on NLI (R. Menon et al., 2022), where a neutral label can be naturally mapped to an abstention. We train `ExEnt-A` on the same tasks as `ExEnt` with the modification of having 'abstain' as an additional class label for each task.

Table 2 shows the results of `TALC-A`, and the baselines when abstention is allowed ('-A' denotes abstention). We find that `TALC-A` achieves the best overall accuracy. More importantly, comparing Table 1 and Table 2, we observe that `TALC` has a smaller drop in performance in comparison to `ExEnt` and `ExEnt-FT` suggesting that `TALC` is better at adapting to multiple teachers even when certain teachers choose to abstain from prediction.

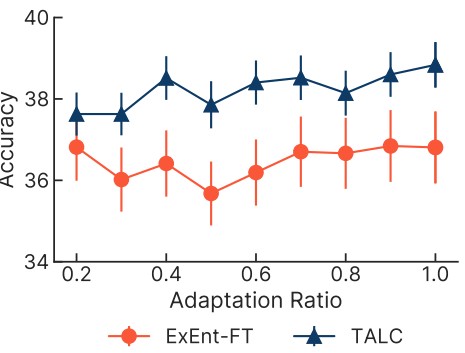

Figure 2: Accuracy (averaged over 6 tasks) of `ExEnt-FT` and `TALC` when training label aggregator with different adaption set sizes. Overall, increasing the adaptation ratio does not impact performance of `ExEnt-FT`, but improves performance of `TALC`.

**Effect of adaptation set size.** We analyze the performance of `ExEnt-FT` and `TALC` by varying size of the adaptation set. Specifically, we vary the adaptation ratio, $\alpha$, from 0.2 to 1.0 (in increments of 0.1) for all six evaluation tasks. [5]

Intuitively, we expect that the accuracy of `ExEnt-FT` and `TALC` to improve with increase in adaptation ratio. However, we empirically observe that the performance of `ExEnt-FT` fluctuates with change in $\alpha$ and does not show a consistent trend of improvement as shown in Figure 2. Meanwhile, as shown in Figure 2, we observe that a larger adaptation set enhances the performance of `TALC` from $37.6\% \rightarrow 38.8\%$ as $\alpha$ increases from $0.2 \rightarrow 1.0$.

**Robustness to number of explanations.** Next, we analyze the robustness of `ExEnt-FT` and `TALC` to changes in the number of explanations provided for adaptation on the new task. We will refer to the fraction of explanations used for adaptation as the

---

[5]The results of `TALC` and `ExEnt-FT` on each individual task with different adaptation ratios can be found in Appendix §G.

| Tasks | Non-Adaptation Baselines | | Adaptation Ratio 0.5 | | Adaptation Ratio 1.0 | |
| --- | --- | --- | --- | --- | --- | --- |
| | ExEnt | ExEnt MV | ExEnt-FT | TALC | ExEnt-FT | TALC |
| banknote-authentication | 46.9 | 48.4 | $45.1_{(0.5)}$ | $49.5_{(0.0)}$ | $45.2_{(0.2)}$ | $\mathbf{49.7}_{(\mathbf{0.3})}$ |
| tic-tac-toe-endgame | 32.8 | 32.3 | $32.3_{(0.0)}$ | $\mathbf{41.1}_{(\mathbf{0.0})}$ | $32.3_{(0.0)}$ | $\mathbf{41.1}_{(\mathbf{0.0})}$ |
| car-evaluation | 10.7 | **17.6** | $4.6_{(2.3)}$ | $14.1_{(3.4)}$ | $3.8_{(0.4)}$ | $16.5_{(0.0)}$ |
| contraceptive-choice | 42.7 | 43.7 | $43.4_{(0.0)}$ | $\mathbf{44.0}_{(\mathbf{0.7})}$ | $43.4_{(0.0)}$ | $43.8_{(0.3)}$ |
| indian-liver-patient | 48.7 | 40.0 | $54.5_{(7.8)}$ | $44.3_{(2.8)}$ | $\mathbf{62.0}_{(\mathbf{4.6})}$ | $47.8_{(0.0)}$ |
| travel-insurance | 31.9 | 33.9 | $\mathbf{34.2}_{(\mathbf{0.0})}$ | $\mathbf{34.2}_{(\mathbf{0.0})}$ | $\mathbf{34.2}_{(\mathbf{0.0})}$ | $\mathbf{34.2}_{(\mathbf{0.0})}$ |
| **Average** | 35.6 | 36.0 | 35.7 | 37.9 | 36.8 | **38.9** |

Table 1: Comparison of zero-shot accuracies (higher is better) between non-adaptation-based `ExEnt` baselines, `ExEnt-FT`, and our proposed method, `TALC`, on the 6 different tasks from `CLUES-Real`. We report the mean and standard deviation for the accuracy across three runs for adaptation-based methods. The numbers in **bold** indicate the best accuracies across methods.

| Tasks | Non-Adaptation Baselines | | Adaptation Methods | |
| --- | --- | --- | --- | --- |
| | ExEnt-A | ExEnt MV-A | ExEnt-FT-A | TALC-A |
| banknote-authentication | 8.0 | 36.7 | $27.1_{(2.5)}$ | $\mathbf{53.9}_{(\mathbf{0.1})}$ |
| tic-tac-toe-endgame | 2.6 | 30.2 | $32.3_{(0.1)}$ | $\mathbf{32.4}_{(\mathbf{0.4})}$ |
| car-evaluation | 2.3 | 2.6 | $\mathbf{14.8}_{(\mathbf{1.8})}$ | $13.3_{(0.8)}$ |
| contraceptive-choice | 22.7 | **32.8** | $19.3_{(2.7)}$ | $31.6_{(1.3)}$ |
| indian-liver-patient | 36.5 | 34.7 | $30.1_{(0.2)}$ | $\mathbf{40.2}_{(\mathbf{0.9})}$ |
| travel-insurance | 15.3 | **24.6** | $15.2_{(2.6)}$ | $23.4_{(0.0)}$ |
| **Average** | 14.6 | 26.9 | 23.1 | **32.5** |

Table 2: Comparison of zero-shot accuracies between `TALC` and the baselines when allowing the `ExEnt` model to abstain from making a prediction (the modified model is denoted as `ExEnt-A`). 'A' stands for 'Abstention' for all the models in the table. For the adaptation methods (`ExEnt-FT`, `TALC`), we report mean and standard deviation across 9 adaptation ratios (0.2 to 1.0). Numbers in **bold** denote the best accuracies across methods.

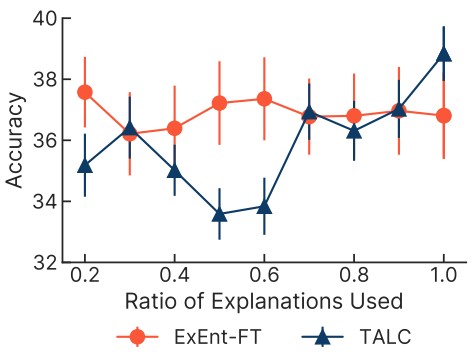

Figure 3: Results for `ExEnt-FT` and `TALC` when varying the number of explanations used for training the label aggregator. Results are averaged over the six evaluation tasks. With increase in number of explanations, the accuracies using `TALC` improve while performance of `ExEnt-FT` is not affected.

explanation ratio $= \frac{\text{\# of available expls}}{\text{\# of all expls}}$. Specifically, we vary the explanation ratio from 0.2 to 1.0, by randomly choosing explanations without replacement, when training the `ExEnt-FT` model and the label aggregator in `TALC`. We keep the adaptation

ratio ($\alpha$) fixed at 1.0 for this analysis.

Figure 3 shows the variation in performance of `ExEnt-FT` and `TALC` with changes in the explanation ratio averaged over the six evaluation tasks. The accuracy of `TALC` drops when increasing the explanation ratio as $0.3 \rightarrow 0.5$, buts shows a consistent increasing trend (from $33.5\% \rightarrow 38.8\%$) when increasing the explanation ratio from $0.5 \rightarrow 1.0$. In contrast, the performance of `ExEnt-FT` fluctuates as the number of available explanations changes. This shows that `TALC` is comparatively more sensitive to the number of explanations used for adaptation.

**Robustness to quality of explanations.** Here we analyze the role of explanation quality on the performance of `TALC`. However, quantifying the quality of explanations in the absence of annotations is a challenging and open research problem. To circumvent this issue, we explore two approaches to quantify explanation quality:

- Individual explanation accuracy: Here, we assume there exists an oracle which has access to all the explanations, the base language guided-

classifier, and the labeled examples. This oracle evaluates the accuracy of each individual explanation of the task by evaluating it on the labeled examples with the base language-guided classifier. We term this accuracy as the individual explanation accuracy and use it a proxy for quantifying the quality of an explanation. For each of the six evaluation tasks, we provide the individual explanation accuracies in Appendix §F.

- Perplexity of an explanation: Assuming access to all labeled examples (needed for the above approach) may be unrealistic for many scenarios. Hence, we also explore a surface-level metric, the perplexity of the explanation, to quantify the quality of an explanation. We obtain perplexity scores for each explanation by using the GPT2-Large pre-trained model (Radford et al., 2019). We provide perplexity scores of each explanation for the six evaluation tasks in Appendix §F.

These aforementioned approaches to quantify the quality of an explanation) can filter out poor quality explanations or selectively choose good quality explanations for adapting the base language-guided classifier. We explore the following scenarios (with adaptation ratio, $\alpha = 1$) to understand the impact of explanation quality:

- Using the top $X$ percentage of explanations: We rank the explanations by accuracy or perplexity for each task and only use the top $X$ percent of the ranked explanations for TALC, where $X = 20, 40, 60, 80, 100$. The results are shown in Figure 4. On average, we observe that TALC performs the best when using only the top 20% of explanations ranked by both accuracy and perplexity. As $X$ increases from $20 \rightarrow 40 \rightarrow 60$, the average performance of TALC decreases, and then keeps increasing. We attribute this trend to the fact that the training of the label aggregator may be sub-optimal with a smaller number of explanations, and improve with more explanations. These results also clearly show that the label aggregator is able to distinguish explanation quality. We note a roughly similar trend when the explanations are ranked by lowest perplexity instead of highest accuracy. This is an encouraging result, and indicates that perplexity of explanations can actually be a reasonable basis for filtering from a large pool of explanations.
- Removing the best explanation: We remove the best (highest accuracy or lowest perplexity) ex-

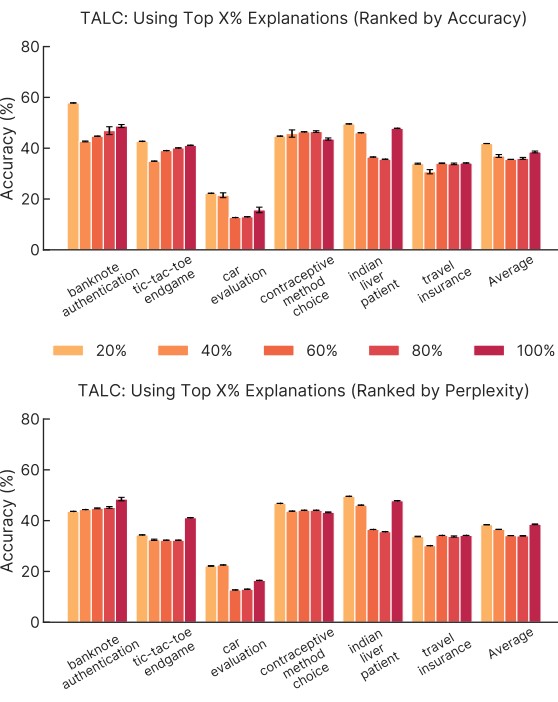

Figure 4: TALC's performance only using the top $X\%$ explanations, where $X = 20, 40, 60, 80, 100$. On average, TALC has the best performance when only using the explanations with the highest quality. The performance of TALC decreases and then increases as we add explanations with lower quality. We see this trend because only the explanations with high quality are used at first and adding explanations with lower quality distract the label aggregator at first, but the label aggregator is able to distinguish high-quality explanations when the number of explanations keeps increasing.

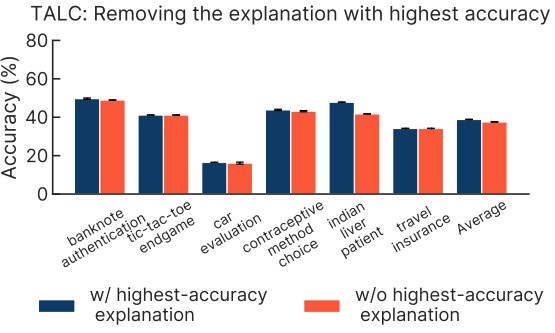

Figure 5: When ranking the explanations by their individual accuracy, removing the best explanation leads to a 1.3% drop in performance on average.

planation from the set of explanations for each the task and adapt TALC. Figure 5 shows that removing the best explanation hurts performance consistently across tasks, as expected. We observe a 1.3% in accuracy drop when ranking the

explanations by accuracy and a 1.0% drop when ranking by perplexity on average across the six tasks (shown in Appendix §I).

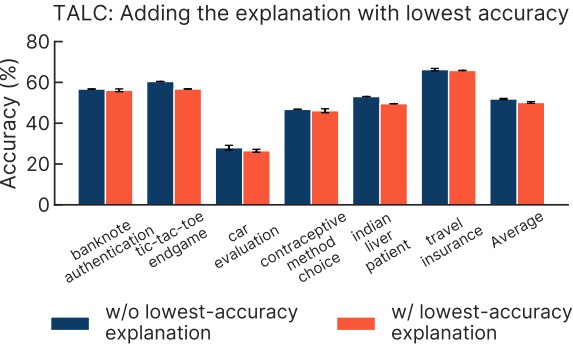

Figure 6: Comparison of TALC's performance before and after adding a low-quality explanation to a set of high-quality explanations. On average, the performance decreases by 1.5% when ranking by accuracy.

- Adding a low-quality explanation to a set of high-quality explanations: Next, we study the impact of low-quality explanations on TALC. For this, we consider two setups. In the first setup TALC utilizes just the top-3 explanations as per their individual accuracies. The individual explanations accuracies can be found in Appendix §F. In the second setup, TALC utilizes the top-3 and the worst explanation (as per individual explanation accuracy) for adaptation. Figure 6 shows the performance of TALC for these settings. When ranking by accuracy, the average decrease in performance due to the addition of low-quality explanation is 1.5%, demonstrating the robustness of TALC to low-quality explanations. We observe a similar trend in results when the explanations are ranking by their perplexity (details in Appendix §I).
- Replacing best explanations with malicious explanations: Next, we create malicious explanations by flipping the labels mentioned by the original explanations. For example, taking the explanation from Figure 1 for the travel-insurance task, we convert '*most college graduates have taken travel insurance*' to '*most college graduates have **not** taken travel insurance*'. We repeat this process for the top-3 explanations ranked by accuracy or perplexity for each of the six evaluation tasks. The results in Figure 7 show a drop in performance of TALC (from 38.8% to 31.5%), as expected, when the top-3 explanations (ranked by their individual accuracies) are modified into

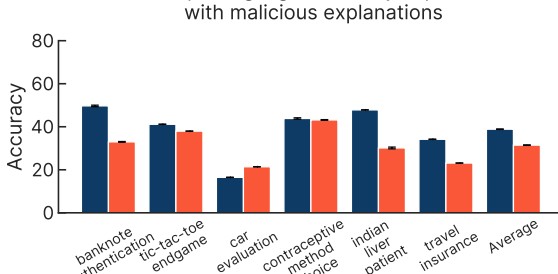

Figure 7: Comparison of TALC's performance before and after replacing good-quality explanations with malicious explanations. On average, TALC's performance drops by 7.3% as good explanations are replaced by malicious explanations.

malicious explanations. When explanations are ranked by perplexity, the results are similar (details in Appendix §I). Surprisingly, for the 'car-evaluation' task, the performance increased from 16.5% to 21.4% on modifying the best explanations to malicious explanations when ranking by accuracy. From the average drop in performance, we can conclude that TALC is susceptible to text-based attacks that may occur through the explanations provided during adaptation. Future work can address the challenge of learning to distinguish between beneficial and adversarial explanations.

**Agnostic nature of TALC w.r.t language-guided classifier.** The flexibility of choosing different models as the underlying language-guided classifiers is an advantage of the TALC framework. The modular design of TALC, i.e, decoupling of (1) how we obtain predictions w.r.t each explanation using a language-guided classifier and (2) how we combine these individual predictions, makes TALC a highly generalizable and flexible framework. To empirically validate this flexibility, we experiment with different LLMs as the underlying language-guided classifier. Table 3 compares the accuracies of TALC and two baselines by using T0-3B (Sanh et al., 2022), OPT-2.7B (Zhang et al., 2022), and Flan-T5-XXL (Chung et al., 2022) as the underlying language-guided classifiers. As baselines, we consider two settings, (1) MV - majority vote of the predictions made by the LLM corresponding to individual explanations and (2) Concat - predicting by considering all explanations (concatenated) together in the context of the LLM. For both set-

| Tasks | T0(3B) | | | OPT(2.7B) | | | FLAN-T5-XXL | | |
|---|---|---|---|---|---|---|---|---|---|
| | Concat | MV | TALC | Concat | MV | TALC | Concat | MV | TALC |
| banknote-authentication | 44.7 | 44.7 | $44.7_{(0.0)}$ | 44.7 | 45.1 | $44.7_{(0.0)}$ | **54.2** | 31.3 | $43.6_{(0.0)}$ |
| tic-tac-toe-endgame | **67.7** | **67.7** | $\mathbf{67.7}_{(0.0)}$ | 32.3 | 49.0 | $64.9_{(1.3)}$ | 32.3 | 40.6 | $56.8_{(0.0)}$ |
| car-evaluation | 3.8 | **71.7** | $\mathbf{71.7}_{(0.0)}$ | 2.3 | 3.8 | $3.8_{(0.0)}$ | 3.8 | 26.6 | $26.6_{(0.0)}$ |
| contraceptive-choice | 23.4 | 23.9 | $23.4_{(0.0)}$ | 27.5 | 33.2 | $33.2_{(0.0)}$ | 38.3 | **45.4** | $\mathbf{45.4}_{(0.0)}$ |
| indian-liver-patient | **67.8** | 33.0 | $48.7_{(0.0)}$ | 32.2 | 32.2 | $32.2_{(0.0)}$ | 66.1 | 48.7 | $56.5_{(0.4)}$ |
| travel-insurance | 38.9 | 65.8 | $65.8_{(0.0)}$ | 42.5 | 65.8 | $65.8_{(0.0)}$ | 71.9 | 71.9 | $\mathbf{72.6}_{(0.4)}$ |
| **Average** | 41.1 | 51.1 | **53.7** | 30.2 | 38.2 | 40.8 | 44.4 | 44.1 | 50.3 |

Table 3: Comparison of accuracies between `TALC` and baselines when using different LLMs as the language-guided classifier on the 6 different tasks from `CLUES-Real`. We report the mean and standard deviation for the accuracy across three runs for adaptation-based methods. The numbers in **bold** indicate the best accuracy across methods.

ting, the prediction is done by prompting the LLM. We provide the prompt templates in Appendix §D. Table 3 shows that `TALC` outperforms both baselines for all of the three LLMs demonstrating the robustness of `TALC` to the choice of the underlying language-guided classifier.

## 5 Discussion & Conclusion

In this paper, we introduce `TALC`, a framework for test-time adaptation of language-guided classifiers that leverages multiple sources of supervision. One conceptual advantage of `TALC` is its agnosticism towards the choice of language-guided classifier, leaving room for future exploration with different models. `TALC` is flexible in terms of what aspects of explanations, teachers and unlabeled examples are used to train the label aggregator. While our approach here trains a label-aggregator for every new task (since our features for the label aggregator include identities of individual explanations), in principle it should be possible to train a unified label aggregator across tasks based on featurized representations of tasks and explanations. Scaling up `TALC` to new datasets with a larger number of tasks would provide valuable insights into its generalizability. Our experiments reveal `TALC`'s susceptibility to malicious attacks and bad-faith actors, which future works can improve on. Despite these challenges, `TALC` suggests exciting opportunities for harnessing the collective wisdom of teachers in real-world applications.

## Limitations

To analyse the impact of quality of an explanation during test-time adaptation, we use individual explanation accuracy as a surrogate measure for its quality in lieu of a standardized metric of explanation quality. However, developing standardized metrics to judge the quality of an explanation remains an open and pressing research challenge.

To analyse robustness of `TALC` w.r.t malicious explanations, we created malicious explanations by flipping the labels mentioned in the best explanation for a task. However, there could be other ways of creating malicious or adversarial explanations, which are more subtle than just flipping a label. For example, one subtle way of altering an existing explanation to a malicious one could be by establishing unwanted correlations between a protected attribute (e.g. gender) and the label for a downstream task (e.g. whether the loan should be approved). Analyzing and improving the robustness of `TALC` to more nuanced adversarial/malicious explanations remains to be explored.

The adapted model obtained by using `TALC` is task dependent, as it uses explanations and unlabeled data specific to the downstream task for adaptation (specifically, for training the label aggregator component). Hence, for every novel task for which we want a adapt a base language-guided classifier, we need access to explanations and unlabeled samples. This requirement (especially obtaining good explanations for adaptation) can be a challenging issue for some real-world scenarios. Improving `TALC` to reduce its dependence on the amount of explanations and/or unlabeled data while still retaining downstream accuracy (post-adaptation) is an interesting direction for future work. The base language-guided classifier used in our experiments, `ExEnt`, is designed to work with a maximum of 512 tokens in its context. Usage of longer context models or even large-scale pre-trained models remains to be explored. The effectiveness of `TALC` under multilingual setting is also unexplored.

## Ethics and Broader Impact

The experiments described in this work are performed over tasks from a publicly available benchmark, CLUES (R. Menon et al., 2022). The data for these tasks do not contain any personally identifiable information. We do not collect or annotate any additional data. For all the experiments in the paper we evaluate using automatic metrics and do not perform any human evaluation.

TALC is agnostic to the base language-guided classifier. We do not foresee major risks with our framework if the inputs provided are appropriate. Like any other natural language guided method there are potential concerns of misguiding a model deliberately by providing erroneous inputs. Measures to detect such bad actors and rectifying erroneous inputs is beyond the scope of this work. However, there is a risk of classifiers perpetuating biases present in the input natural language explanations (for example, some explanations may describe the label in terms of sensitive or inappropriate features). Biased or discriminatory explanations can result in biased predictions and contribute to unjust outcomes.

The broader impact of this work can lead to development of frameworks that enable efficient adaptation of AI systems. Developing language-guided adaptable systems can improve the impact and usability of AI systems in daily life, especially on the long tail of tasks with limited labeled data. However, the responsible development and deployment of these models would require domain-specific expertise, involving collaboration with experts and stakeholders to understand the implications and ensure ethical considerations are met. Close attention should be paid to the specific contexts in which the classifiers are applied to minimize negative consequences and maximize positive impacts.

## Acknowledgments

The authors would like to thank the anonymous reviewers for their suggestions and feedback on the work. This work was supported in part by NSF grant DRL2112635. The views contained in this article are those of the authors and not of the funding agency.

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

# Appendix

## A  Usage of `TALC`

1. An example of real-world cases where the entire set of test samples can be realistically accessed:

   Let's consider the case of a product category classifier for products on the Amazon database. In this case, developers will first define classifiers using some training data and deploy the classifier on the entire database to label examples.

2. Method to use TALC when test samples are observed one-by-one:

   Even if we do not have access to the entire test set and the classifier observes unlabeled samples one by one, TALC can be deployed in practice as:

   (a) For a predetermined number of samples, the language-guided classifier is deployed off-the-shelf (Note: In this work, this would be the same as using `ExEnt` for those samples).
   (b) The aforementioned samples can now be pooled together as an adaptation set, and we can adapt the language-guided classifier using `TALC`.

   In other words, we incur a "warm-up" phase, where the un-adapted classifier is used, following which we adapt the classifier using TALC by considering the set of samples observed during warm-up as an adaptation set.

## B  Details of evaluation tasks

We use 6 real world classification tasks from R. Menon et al. (2022) as our evaluation tasks. The tasks considered are – uci/banknote-authentication, uci/tic-tac-toe-endgame, uci/car-evaluation, uci/contraceptive-method-choice, uci/indian-liver-patient, and kaggle/travel-insurance. Examples of these tasks can be found at the CLUES website with the following link: https://clues-benchmark.github.io. Among the above tasks, uci/car-evaluation and uci/contraceptive-method-choice are multi-class classification tasks while the rest tasks are binary classification tasks. The numbers of examples in test set of each task are 275, 195, 346, 295, 115, 398 for uci/banknote-authentication,

| Models | Prompt |
|---|---|
| T0 FLAN-T5 | Explanations: <explanation_1>. <explanation_2>. …. <explanation_n> Note Details: <feat_1> equal to <feat_1_value>. <feat_2> equal to <feat_2_value> …From the explanations, is the note fake or original? Answer: |
| OPT | The following is a classification task that uses the following explanations. Based on the explanation classify the subsequent sample:

Explanations:
- <explanation_1>
- <explanation_2>
⋮
- <explanation_n>

Note Details: <feat_1> equal to <feat_1_value>. <feat_2> equal to <feat_2_value> …
From the explanations, is the note fake or original?

Answer: |

Table 4: Prompt templates used for large language model experiments in 4.4.

uci/tic-tac-toe-endgame, uci/car-evaluation, uci/contraceptive-method-choice, uci/indian-liver-patient, and kaggle/travel-insurance respectively.

## C  Hyperparameter and Compute Details

We train the `ExEnt` model following the hyperparameters in R. Menon et al. (2022), e.g. a learning rate of 1e-5 for 5 epochs, batch size of 2, and evaluation batch size of 16. For the label aggregator training, we did hyper-parameter search for each of the tasks and report the best hyper-parameters in Table 5.

For fine-tuning the `ExEnt` model, compute time ranged from 1 hr for the shortest jobs with smaller data sizes to 2 hours on 1 RTX 2080Ti GPU. For fine-tuning the label aggregator, compute time is within 1 minute.

## D  Prompt Templates for LLM Experiments

For the experiments using large language models, we used the prompts elaborated in Table 4.

| TASK | LEARNING RATE | EPOCH |
|------|---------------|-------|
| banknote-authentication | $4.38e-08$ | 500 |
| tic-tac-toe-endgame | $7.31e-08$ | 40 |
| car-evaluation | $2.48e-04$ | 500 |
| contraceptive-method-choice | $5.53e-03$ | 100 |
| indian-liver-patient | $7.91e-03$ | 50 |
| travel-insurance | $7.31e-08$ | 40 |

Table 5: The best hyper-parameters for training label aggregator.

## E  Learned Label Aggregator Explanation Weight

We analyze the learned values of the weights in the label aggregator, $\mathcal{L}_w^{\text{agg}}$, to interpret the contribution of each explanation towards the final prediction of `TALC` at an adaptation ratio of 1.0. First, we calculate the accuracy of each individual explanation of a task by using it with `ExEnt` for classification on the entire test set. These individual explanation accuracies serve as a proxy for their relative quality. The visualization for the 6 datasets' learned explanation weights of label aggregators and the learned explanation weight trends are shown in Figure 9.

Here, we also show the average learned explanation weight for explanations with/without quantifiers and the average learned explanation weight for explanations with/without conjunctions in Table 6. Quantifiers are words like 'always' and 'usually'. Conjunctions are words like 'and' and 'or'. We use the same quantifier and conjunction words following R. Menon et al. (2022)

| Dataset | Quantifier | No Quantifier | Conjunction | No Conjunction |
|---------|-----------|---------------|-------------|----------------|
| banknote authentication | 0.18 | 0.14 | - | 0.14 |
| tic-tac-toe endgame | 0.20 | 0.12 | 0.29 | 0.16 |
| car evaluation | 0.01 | 0.18 | 0.15 | 0.18 |
| contraceptive choice | 0.15 | 0.21 | 0.14 | 0.22 |
| indian-liver patient | - | 0.51 | 0.33 | 0.57 |
| travel insurance | 0.18 | 0.17 | 0.16 | 0.18 |
| **Average** | 0.14 | 0.22 | 0.21 | 0.24 |

Table 6: The average learned explanation weight for explanations w/wo quantifiers and explanations w/wo conjunctions of label aggregators for each task. Empty values in the table indicate that the linguistic element was absent in the explanations for the corresponding dataset.

## F  Individual Explanation for Each Task

We show all the available natural language explanations for the six `CLUES-Real` dataset we use in this paper in Table 8 to 13. In Table 8 to 13, We also report the accuracy when using only one explanation at a time with `ExEnt` and the perplexity of each explanations. In Figure 10, we analyze the correlation between accuracy and perplexity of all explanations. There is a positive correlation between accuracy and perplexity of all the explanations.

## G  Results for Models Without Abstention

Here, we show `ExEnt-FT` and `TALC` experiment results without abstention on different adaptation size for each 6 tasks from `CLUES-Real` in Figure 11.

## H  Few Shot Learning

Here, we run a few-shot supervised version of `ExEnt`. We fine-tune the `ExEnt` model using $k$ samples with gold labels from the evaluation tasks, where $k = 4, 8, 16, 32$. We report the results in Table 7. We observed that the test accuracy of this few-shot trained `ExEnt` is better than `TALC`. The performance with a few-shot model is better as the gold labels are quite different from noisy aggregated labels used by `TALC` for adaptation. We observed a huge label imbalance in the intermediate `ExEnt` model that results in lower accuracy for both `TALC` and `ExEnt-FT`, both of which leverage `ExEnt` 's predictions as noisy pseudo labels.

| Dataset | $k=4$ | $k=8$ | $k=16$ | $k=32$ |
|---------|-------|-------|--------|--------|
| banknote authentication | $47.5_{(2.7)}$ | $51.2_{(1.5)}$ | $54.9_{(5.2)}$ | $53.6_{(4.1)}$ |
| tic-tac-toe endgame | $55.6_{(8.1)}$ | $59.1_{(6.5)}$ | $63.7_{(2.8)}$ | $64.2_{(1.8)}$ |
| car evaluation | $61.7_{(6.2)}$ | $52.6_{(1.6)}$ | $68.1_{(1.4)}$ | $69.4_{(2.3)}$ |
| contraceptive choice | $34.4_{(4.3)}$ | $32.9_{(4.4)}$ | $36.8_{(0.2)}$ | $40.0_{(2.3)}$ |
| indian-liver patient | $66.9_{(2.6)}$ | $67.4_{(1.3)}$ | $63.8_{(3.4)}$ | $67.5_{(0.8)}$ |
| travel insurance | $53.9_{(10.3)}$ | $57.0_{(7.1)}$ | $53.9_{(1.7)}$ | $62.5_{(3.7)}$ |
| **Average** | 53.3 | 53.4 | 56.9 | 59.5 |

Table 7: Few-shot fine-tuning with `ExEnt`. We report the mean and standard deviation for the accuracy across three runs using different seeds.

We would like to emphasize that a few-shot supervised model is **not** an ideal baseline for our framework. This is because `TALC` is designed to work as an unsupervised approach for test-time adaptation. Hence, few-shot fine-tuning would represent an upper bound for `TALC`. Alternatively, few-shot finetuning can be treated as a comple-

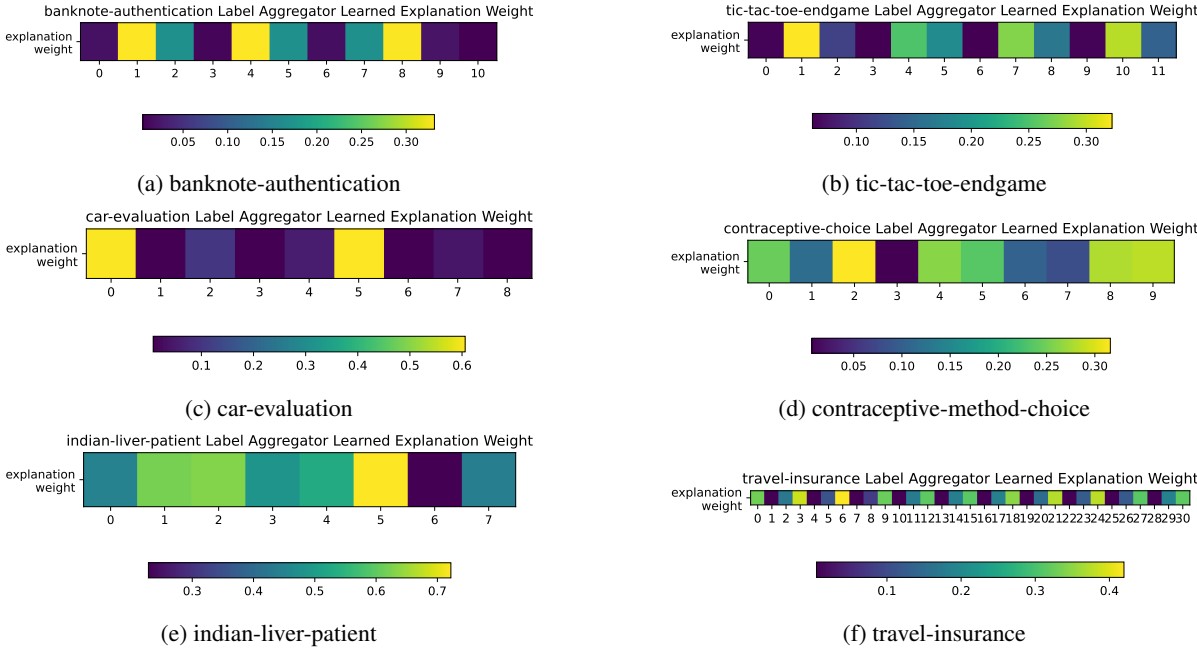

Figure 8: Learned label aggregator accuracy factor for the 6 `CLUES-Real` evaluation datasets used in our work.

mentary approach to `TALC` and can be paired with `TALC` for real-world test time adaptation scenarios. Combining these two paradigms for improved test time adaptation is an interesting direction for future work and is beyond the scope of this paper.

## I   Ranking by Perplexity

We show the results of the ablations studies described in Section 4.4 ranking by perplexity here in Figure 12 to 14.

For the setting where we add one low-quality explanation to a set of high-quality explanation, when ranking by perplexity, the average performance increases by 0.6%, caused by the performance increase of the banknote-authentication task while the other tasks' performance either decreases or stays the same. According to Table 8, the added explanation for banknote-authentication has the highest perplexity and the second highest accuracy, suggesting the accuracy metric may have stronger impact to the performance than perplexity does.

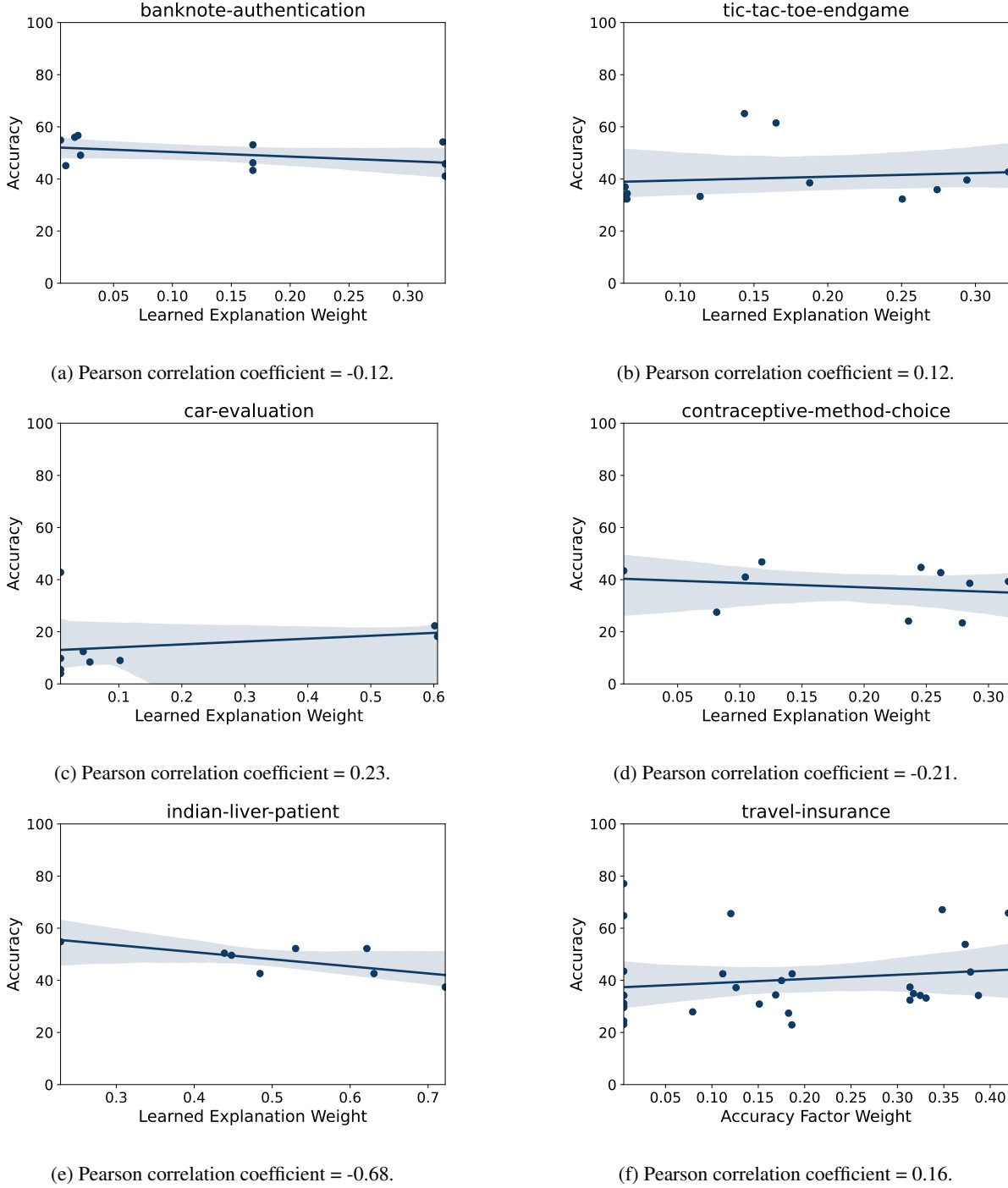

Figure 9: Relationship between learned accuracy factor weight and the corresponding explanations' accuracy for (a) banknote-authentication, (b) tic-tac-toe-endgame, (c) car-evaluation, (d) contraceptive-method-choice, (e) indian-liver-patient, and (f) travel-insurance.

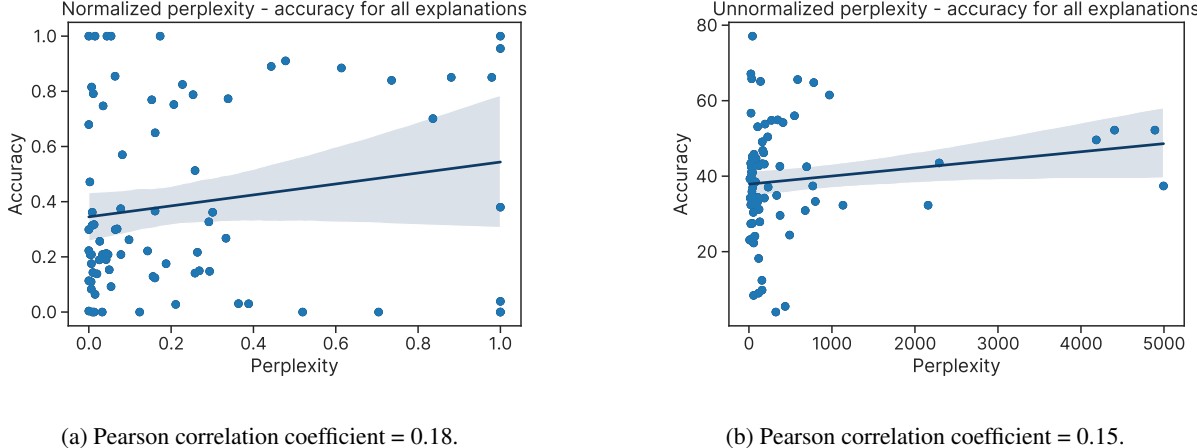

(a) Pearson correlation coefficient = 0.18.                (b) Pearson correlation coefficient = 0.15.

Figure 10: Relationship between perplexity and accuracy of explanations: (a) Normalized, (b) Un-normalized

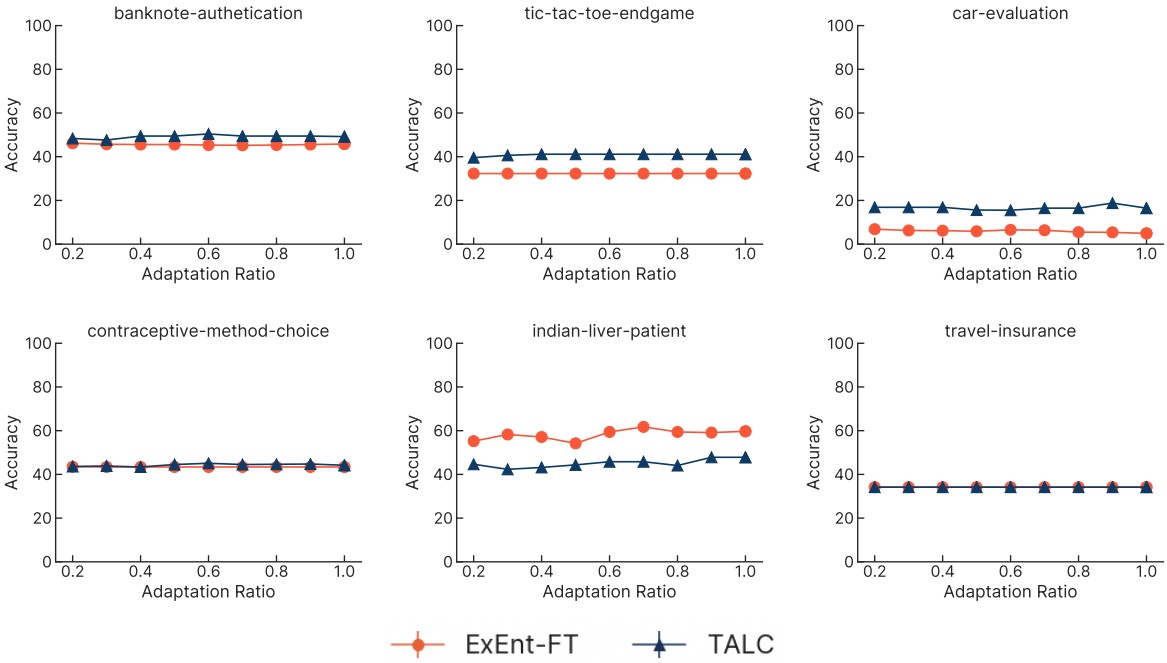

Figure 11: Performance of `ExEnt-FT` and `TALC` with different adaptation ratios for each of the 6 evaluation tasks.

| No. | Explanation | Accuracy | Perplexity |
|---|---|---|---|
| 0 | If the variance of the note is a negative number, it's more likely to be an original note. | 56.7% | 25.05 |
| 1 | If the kurtosis of the bill is a negative number, it's more likely to be a fake note. | 41.1% | 31.82 |
| 2 | A fake banknote has a variance level over 4.0. | 46.2% | 177.85 |
| 3 | Notes that have a negative Variance level will also have a negative Entropy level. | 45.1% | 39.02 |
| 4 | Notes that have a positive Skewness level will have a negative Kurtosis level. | 45.8% | 61.06 |
| 5 | Variance above 4 leads to the results of FAKE. | 43.3% | 160.22 |
| 6 | Below 3.80 skewness leads to the original. | 56.0% | 548.93 |
| 7 | Variance above 1.00 leads to the FAKE. | 53.1% | 105.15 |
| 8 | Below the 3.75 Sketwness leads to the ORIGINAL. | 54.2% | 410.40 |
| 9 | Entropy is low value so it is fake. | 49.1% | 160.25 |
| 10 | Kurtosis is high value so it is original. | 54.9% | 346.58 |

Table 8: All explanations for banknote-authentication task used in this paper.

| No. | Explanation | Accuracy | Perplexity |
|---|---|---|---|
| 0 | The game is usually not over yet when there are at least two blank squares. | 34.4% | 56.82 |
| 1 | The game is usually over when the players have taken all the middle squares. | 42.7% | 51.84 |
| 2 | The top middle square results in x winning. | 33.3% | 800.38 |
| 3 | The bottom middle square results in x winning. | 32.3% | 1132.84 |
| 4 | When the middle-right-square is left blank, x is less likely to win. | 32.3% | 94.01 |
| 5 | Whoever marks the top-left-square is unlikely to win. | 38.5% | 78.83 |
| 6 | The player who moves second is much more likely to lose if they place in the middle-left-square. | 37.0% | 45.58 |
| 7 | The middle-right-square is the square that is most likely to go unused during a game. | 35.9% | 36.20 |
| 8 | Top left square X indicates the Negative group. | 61.5% | 969.94 |
| 9 | Without b categories in middle middle square comes under the Positive group. | 32.3% | 2158.20 |
| 10 | An O in both the top-left and bottom-right is likely to be positive. | 39.6% | 24.43 |
| 11 | A blank in the middle-right will likely lead to negative. | 65.1% | 139.76 |

Table 9: All explanations for tic-tac-toe-endgame task used in this paper.

| No. | Explanation | Accuracy | Perplexity |
|---|---|---|---|
| 0 | If safety is high, then the car will not be unacceptable. | 22.3% | 56.95 |
| 1 | If maintenance cost is medium, then the car will not be unacceptable. | 42.8% | 121.90 |
| 2 | A capicity for 4 or more persons makes the vehicle acceptable for resale. | 9.0% | 115.36 |
| 3 | High safety ratings generally make vehicles acceptable for resale. | 9.8% | 158.10 |
| 4 | A low buying cost is a good indicator of a vehicle being acceptable for resale. | 8.4% | 55.96 |
| 5 | Most people having the passenger capacity of 4 or more have good acceptability for car resale. | 18.2% | 117.22 |
| 6 | Cars with low buying and maintenance cost are highly acceptable for resale | 5.5% | 436.58 |
| 7 | Cars with higher safety and capacity are highly acceptable for resale. | 12.4% | 156.38 |
| 8 | Cars with higher safety and medium luggage boot size are highly acceptable for resale. | 4.0% | 323.92 |

Table 10: All explanations for car-evaluation task used in this paper.

| No. | Explanation | Accuracy | Perplexity |
|-----|-------------|----------|------------|
| 0 | A husband's education has a high chance of a long-term contraceptive being used. | 44.7% | 85.88 |
| 1 | A wife's education level usually determines if a long term contraceptive was used. | 46.8% | 166.37 |
| 2 | Women with low education are more likely to use short-term contraception. | 39.3% | 12.18 |
| 3 | Women with high education age 40 and under are more likely to use long-term contraception. | 43.4% | 22.01 |
| 4 | Women with two or more children used short and long term methods. | 42.7% | 47.27 |
| 5 | The least educated women either used short-term method or didn't use any contraceptive method. | 24.1% | 72.05 |
| 6 | If the wife's education is high, then the contraceptive method used is long-term. | 41.0% | 44.04 |
| 7 | If the wife's education is not high, then the contraceptive method used is no-use or short-term. | 27.5% | 41.12 |
| 8 | If the wife's education is not high, then the contraceptive method used is not long-term. | 23.4% | 31.23 |
| 9 | If the wife's education is not high, then the contraceptive method is short-term. | 38.6% | 37.03 |

Table 11: All explanations for contraceptive-method-choice task used in this paper.

| No. | Explanation | Accuracy | Perplexity |
|-----|-------------|----------|------------|
| 0 | The SGPT High percentage so the liver patient was yes. | 49.6% | 4184.78 |
| 1 | The SGPT Low percentage so the liver patient was no. | 52.2% | 4891.02 |
| 2 | Patients over the age of forty are liver patients. | 42.6% | 54.92 |
| 3 | Patients who has SGOT range greater than forty are liver patients. | 42.6% | 373.03 |
| 4 | Decreased SGPT Value ensures no liver patient. | 52.2% | 4404.52 |
| 5 | Age group above 40 ensures liver patient. | 37.4% | 4994.60 |
| 6 | Some people have more age and have the SGOT and they are liver patient. | 54.8% | 272.55 |
| 7 | Some people have minimum age and they are liver patient. They are somewhat accurate. | 50.4% | 226.11 |

Table 12: All explanations for indian-liver-patient task used in this paper.

| No. | Explanation | Accuracy | Perplexity |
|---|---|---|---|
| 0 | Frequent flyers with an annual income over 1 million usually take travel insurance. | 33.2% | 104.36 |
| 1 | People with an annual income over 1 million and under 30 years old usually take travel insurance. | 30.4% | 53.40 |
| 2 | Travelers who are 29 and older take traveler insurance. | 34.4% | 105.36 |
| 3 | Travelers who have not traveled abroad before are more likely to take traveler insurance. | 34.2% | 22.74 |
| 4 | Frequent flyer travelers with an annual income above 1 million frequently take travel insurance. | 31.2% | 121.70 |
| 5 | Travelers older than 25 years old and with an income below 1 million do not usually take travel insurance. | 42.5% | 27.45 |
| 6 | Most passengers who are not frequent fliers do not use travel insurance. | 65.8% 33.28 | |
| 7 | Most passengers who have not traveled abroad do not use travel insurance. | 77.1% | 42.87 |
| 8 | Most passenger with an income higher than 100,000 use travel insurance. | 27.9% | 131.69 |
| 9 | People who have never traveled abroad before are more likely to have taken travel insurance. | 34.2% | 15.95 |
| 10 | People with an annual income below 1,000,000 are less likely to have traveled abroad than those with annual incomes above 1,000,000. | 23.1% | 8.36 |
| 11 | People with an annual income above 1,000,000 are more likely to have taken travel insurance. | 39.9% | 26.63 |
| 12 | More frequent flyers have taken travel insurance. | 34.9% | 334.32 |
| 13 | More people that travel abroad have taken travel insurance. | 34.2% | 114.91 |
| 14 | People who are non-frequent flyers and are college graduates are less likely to get travel insurance. | 27.4% | 22.28 |
| 15 | People who make a million or more and are frequent fliers are more likely to get travel insurance. | 32.4% | 22.73 |
| 16 | Most people who didn't travel abroad before had taken Travel Insurance. | 34.2% | 84.50 |
| 17 | Most college graduate have taken Travel Insurance. | 64.8% | 780.98 |
| 18 | Those who are college graduates and in their 20s are somewhat likely to purchase travel insurance. | 22.9% | 28.62 |
| 19 | Those who have never travelled abroad and are not frequent flyers often do not purchase travel insurance. | 67.1% | 23.85 |
| 20 | Travelled Abroad Before "No" categories indicates the "No" Travel Insurance Taken. | 24.4% | 490.45 |
| 21 | College Graduate "Yes" categories leads to the "Yes" Travel Insurance Taken. | 30.9% | 678.44 |
| 22 | Frequent Flyer "No" indicates the "No" Travel Insurance Taken. | 53.8% | 194.31 |
| 23 | Annual income categories above 1049999 indicates the "Yes" Travel Insurance Taken. | 43.5% | 2291.89 |
| 24 | Most college graduates that make more than 1000000 annually have taken travel insurance. | 37.2% | 231.38 |
| 25 | Most college graduates from 26 to 34 have taken travel insurance. | 43.2% | 185.20 |
| 26 | About have of college graduates that have not travelled abroad before have taken travel insurance. | 34.2% | 186.52 |
| 27 | Frequently flyer "No" categories indicates the "No" travel insurance taken group. | 65.6% | 586.77 |
| 28 | Annual income above 1049999 indicate the "Yes" travel insurance taken group. | 37.4% | 769.28 |
| 29 | Frequent flyer "No" categories indicates the "No" travel insurance taken group. | 29.6% | 375.25 |
| 30 | Annual income above 1049999 indicates the "Yes" travel insurance taken group. | 42.5% | 695.34 |

Table 13: All explanations for travel-insurance task used in this paper.

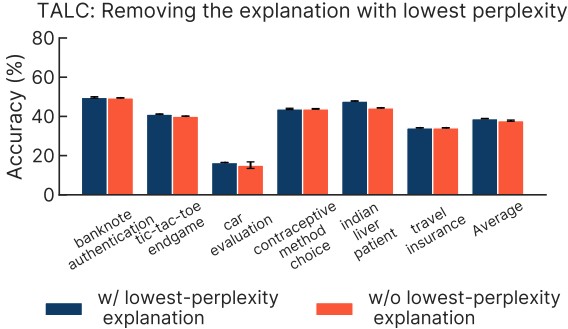

Figure 12: When ranking the explanations by their individual perplexity, removing the best explanation leads to a 1.0% drop in performance on average.

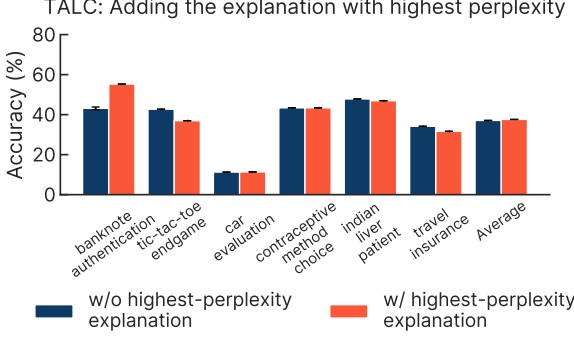

Figure 13: Comparison of TALC's performance before and after adding a low-quality explanation to a set of high-quality explanations. On average, the performance increases by 0.6% when ranking by perplexity.

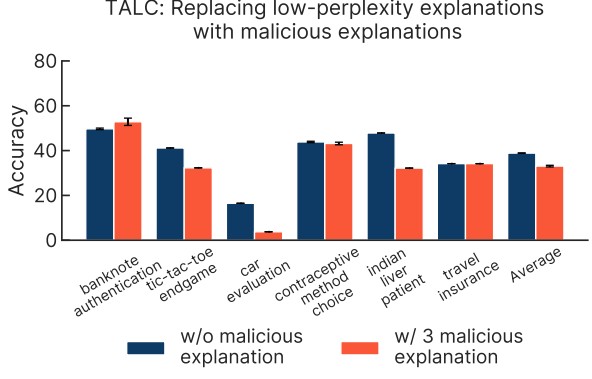

Figure 14: Comparison of TALC's performance before and after replacing good-quality explanations with malicious explanations. On average, TALC's performance drops by 5.6% as good explanations are replaced by malicious explanations.