# OpenReview forum: "Leveraging Multiple Teachers for Test-Time Adaptation of Language-Guided Classifiers"
_EMNLP/2023/Conference — EMNLP 2023 Findings_

### Official Review · Reviewer_382n · 2023-07-30

**Soundness:** 4

**Excitement:**

3: Ambivalent: It has merits (e.g., it reports state-of-the-art results, the idea is nice), but there are key weaknesses (e.g., it describes incremental work), and it can significantly benefit from another round of revision. However, I won't object to accepting it if my co-reviewers champion it.

**Paper Topic And Main Contributions:**

Language-guided classifiers refers to the problem of classifying $n$ test examples when given $m$ natural language explanations describing the classification task.
This paper proposes TALC: Test-time Adaptation of Language-guided classifiers, which extends from prior work ExEnt (R. Menon et al., 2022) and includes a new label aggregation component for test time adaptation.
In ExEnt, the predictions are made by mean-pooling results of running $m$ NLI predictions (whether each of the $m$ explanation entails the classification result).
In TALC, the label aggregator has learnable parameters $w$ which weight the predictions from the $m$ explanations. $w$ is learned on unlabeled test examples using EM algorithm.
On 6 tasks in the CLUES benchmark, TALC achieves strong empirical performance (outperforms ExEnt by 3.3%). Additional analysis is done regarding abstention, effect of adaptation-set size, number of explanations and quality of explanations.

**Questions For The Authors:**

* One hypothesis of TALC is that it assigns "higher weightage to high-quality explanations" (Line 344). However in Figure 9 it seems that the learned weight is not having strong positive correlation with the explanation's quality (reflected by accuracy). Could you please elaborate on this observation?
* Line 375: Is TALC-A another method or is it the same as the TALC method in Table 1? From my understanding TALC already takes abstention into account.
* Table 2: What's the reasoning behind reporting "mean and standard deviation across 9 different adaptation ratios"?

**Reasons To Accept:**

* Training language-guided classifiers is an important research topic. It is a unique learning paradigm as opposed to supervised learning and is very interesting given the rise of large language models.
* TALC has strong empirical results, outperforming ExEnt (previous state-of-the-art method) by 3.3% on average.
* Extensive analysis on the proposed TALC framework under different settings.

**Reasons To Reject:**

* Need more justification regarding technical contributions and novelty: As mentioned in the related work section (data programming) and method introduction (Line 205,262), the work is closely related to existing data programming technique such as Snorkel-MeTaL. MeTaL proposes to "learn a re-weighted model of labeling functions". For now I cannot clearly distinguish the difference between TALC and MeTaL and would request more justification on this from the authors.
* Need baselines with larger language models: As stated in the abstract, large models such as GPT-3 are good at learning from explanations and instructions. It would be extremely helpful to provide the baselines on these larger models for the language-guided classifier task. In addition, since TALC is an flexible framework (Line 543), would it be possible to apply TALC when these larger models are used to produce the $M\in\mathbb{R}^{n\times m}$ matrix?
* Need direct/clear takeaways from the analysis. Sec 4.4 provides comprehensive analysis on TALC's performance under different scenarios. I appreciate these extensive efforts. However, it is not very clear how these analysis can be used to guide the application of TALC. For example, should one always use adaptation ratio $\alpha=1$? Given Figure 3, how many explanations is sufficient for TALC to work well?

**Reproducibility:**

3: Could reproduce the results with some difficulty. The settings of parameters are underspecified or subjectively determined; the training/evaluation data are not widely available.

**Reviewer Confidence:**

3: Pretty sure, but there's a chance I missed something. Although I have a good feel for this area in general, I did not carefully check the paper's details, e.g., the math, experimental design, or novelty.

**Typos Grammar Style And Presentation Improvements:**

* Line 252: It may be helpful to provide more description on the EM algorithm used to learn $w$.

---

> ### Author Rebuttal · Authors · 2023-08-28
>
> Thank you for the thorough review! We are encouraged by the fact that the reviewer acknowledges the importance of the research problem addressed in our work and the superior performance of our framework against the baselines. Below we address some of the concerns raised by the reviewer:
>
> * **Technical contribution**: Traditional data programming techniques require access to labeling functions (essentially semantic parsers that are either learned or provided as inputs). On the other hand, TALC does not require any such labeling function/semantic parsers. TALC focuses on adapting *any* language guided classifier during test-time. The underlying language guided classifier plays the role of labeling function by assigning pseudo labels while being more generalizable and robust than traditional labeling functions (since the language-guided classifier can use natural language explanations directly rather than parsing them into a specific format).\
> Our framework, TALC, provides the first proof-of-concept in leveraging data programming to adapt any language guided classifier during test-time. TALC is agnostic to the choice of language-guided classifier or the label aggregation algorithm. Through specific choices of using ExEnt and Metal as the classifier and label aggregation algorithm we empirically show the efficacy of TALC. Our experiments also highlight the robustness of TALC with the number and quality of explanations and with test-time adaptation set size.
> Exploring other alternatives of classifiers and label aggregation techniques is beyond the scope of this work and can be a separate research avenue in itself, which we leave for future work.
>
>
> * **Baselines with Large Language Models**: Larger language models can certainly be applied as the language-guided classifier of the framework (i.e, one could use any LLM in place of ExEnt under the same TALC framework). This flexibility of choosing different models as the underlying language-guided classifiers is an advantage of the TALC framework that we advocate.
>
> * **Takeaways from Analysis**: Thanks for the suggestion! We summarize the takeaways from Section 4.4 more concisely below and will ensure to add these takeaways to the final version of the draft:
>     - If we have enough compute, α=1 is the best adaptation ratio to utilize for test-time adaptation, i.e. utilize the entire set of available test data to adapt.
>     - Given the sensitivity of TALC to the number of explanations, it is best to always utilize all available explanations for a task during adaptation (Figure 3). If you have a metric (such as, perplexity of the explanation) to compute the quality of explanations, however, utilizing the top 20% of the available explanations as per the metric is also a viable option to reduce computation (Figure 4).
>     - From Figure 6, if the user is privy to the fact that an explanation is malicious, then they should remove such explanations as the performance of the adapted classifier can be drastically impacted by such explanations.
> * **TALC assigns higher weightage to higher-quality explanations, but there is no strong positive correlation**:  We would like to clarify that we do not mention the association between explanation accuracy and weightage to be *strong* positive correlation in the draft. As rightly pointed out by the reviewer, these weights indicate a mild positive correlation. We conjecture that this lack of *strong* correlation could be attributed to factors other than accuracy, such as perplexity of explanations, that also affects the weightage assigned by the label aggregator of TALC. As noted in R. Menon et al. (2022), ExEnt (which is the underlying language-guided classifier of TALC) does not handle explanations with conjunctions, negations, or quantifiers well and hence that potentially affects the explanation rankings as well.
>
> * **TALC-A v.s TALC**: While in principle TALC can handle abstention, the usage of ExEnt as the language-guided classifier was a limitation as ExEnt was not trained to abstain from prediction. Hence, to analyze efficacy of TALC under abstention, we consider a modified version of ExEnt, namely ExEnt-A which can now abstain from prediction. We do this by re-directing the neutral logit from NLI step to a separate “abstain” label. More details about this can be found in L#356-383. The version of TALC which uses this modified language-guided classifier, ExEnt-A, is referred as TALC-A in our paper.
>
> * **Why report mean, std over different adaptation ratios?** The aim of Table 2 is to analyze the performance of different approaches when models can abstain from classification. In table 2, we provide a concise statistic that captures performance of TALC and baselines across a range of adaptation set sizes by computing the mean and standard deviation over the nine adaptation ratios (thus reflecting small to large sizes of adaptation set).

---

### Official Review · Reviewer_7rv9 · 2023-08-03

**Soundness:** 3

**Excitement:**

3: Ambivalent: It has merits (e.g., it reports state-of-the-art results, the idea is nice), but there are key weaknesses (e.g., it describes incremental work), and it can significantly benefit from another round of revision. However, I won't object to accepting it if my co-reviewers champion it.

**Paper Topic And Main Contributions:**

This paper proposes a framework for classification of unlabeled data, tailored to an scenario in which both (i) a set of natural language explanations from multiple sources and (ii) a set of unlabeled data examples are available. Specifically, it makes use of the data programming paradigm of Ratner et al. (2016) to train a function which aggregates the explanation-dependent predictions of a base language-guided classifier. The latter is chosen to be the ExEnt model of Menon et al. (2022).

The authors empirically showed that their method generally outperforms the baselines. The authors also included an interesting analysis of the sensibility of their method to the number and quality of the available explanations.


*Links to references*

- Ratner et al (2016): https://proceedings.neurips.cc/paper/2016/file/6709e8d64a5f47269ed5cea9f625f7ab-Paper.pdf

- Menon et al. (2022): https://aclanthology.org/2022.acl-long.451/

**Questions For The Authors:**

I will just copy here the question from above:

1. Why is it sensible to model the joint distribution P(X, E, Y) with Eq. 1?

2. Why to use EM as opposite to variational inference?

3. What are the disadvantages of using MAP estimates in this problem?

4. Why didn't you consider a different baseline (i.e. one not based on the ExEnt model)?

**Reasons To Accept:**

- The paper tackles the problem of test-time adaptation for unlabeled classification which, as pointed out by the authors, aligns well with real-world scenarios.

- According to the authors, their methodology represents a first application of the data programming paradigm to test-time adaptation for unlabeled classification.

- The authors studied the sensibility of their methodology to plausible perturbations.

**Reasons To Reject:**

- The main components of the methodology proposed by the authors could definitely be explained better. It feels the authors assume every reader is well acquainted with both, the data programming paradigm of Ratner et al. (2016) and the ExEnt model of Menon et al. (2022). The label aggregator, for example, represents the heart of the method. Yet it's not very easy to follow the description provided by the authors.

Why is it sensible to model the joint distribution P(X, E, Y) with Eq. 1?; Why to use EM as opposite to variational inference?; What are the disadvantages of using MAP estimates in this problem?

None of these points are discussed.

Could you include, be in the Appendix or in Algorithm 1, the update EM equations?

- The baselines are all modifications of the language-guided classifier of Menon et al. (2022). The proposed methodology also leverages this language-guided classifier. It's therefore not difficult to believe that the proposed methodology outperforms the baselines, since the former builds on the latter. Why didn't you consider a different baseline?

[Note: I am not knowledgeable about the problem of test-time adaptation for unlabeled classification.  I understand that it might be that no other method has been developed for this problem in the past, and hence that comparing with another baseline is difficult. It'd be worth elaborating on this, if this were to be the case]

- No code/repo was provided (or at least I can't find it). This, together with the first point above, makes it very difficult to both understand the method and reproduce any of the reported results.

**Reproducibility:**

3: Could reproduce the results with some difficulty. The settings of parameters are underspecified or subjectively determined; the training/evaluation data are not widely available.

**Reviewer Confidence:**

2: Willing to defend my evaluation, but it is fairly likely that I missed some details, didn't understand some central points, or can't be sure about the novelty of the work.

**Typos Grammar Style And Presentation Improvements:**

- Typo. Lines 342: aggregator **in** results

- Typo. Lines 549: in **principal**

- Could you include, be in the Appendix or in Algorithm 1, the update EM equations?

---

> ### Author Rebuttal · Authors · 2023-08-28
>
> Thank you for the thorough review! We are encouraged by the fact that the reviewer recognizes the relevance of the issue we address and how it relates to real-world situations. Additionally, we appreciate the acknowledgment of our sensitivity analysis. Below we address some of the concerns raised by the reviewer:
>
> * **Why is it sensible to model P(X, E, Y) ?** The observations in this case are the unlabeled instances X and the explanations E. Thus, we would like to infer the latent labels Y that would maximize the probability of these observations, i.e. P(X, E). For this, the natural choice would be to model the joint probability P(X, E, Y) and marginalize out Y.
>
> * **Why use EM over VI?** We need to not only infer the latent labels Y for the unlabeled data, but also learn the model parameters for the graphical aggregator (which is why we need both the inference in the E step and the learning in the M step). If the reviewer’s question is why we don’t use a variational inference approach in the E-step (rather than a sampling-based approach), this was simply because of ease of implementation (the Gibbs sampling updates for this specific case are really simple). It is possible that a VI based approach might work better.
>
> * **Disadvantages of MAP estimates**: MAP normally works better when datasets are small. In our case, the test set we used from CLUES normally has around 200 to 300 examples, which makes MAP suitable for the tasks.
> As for situations where MAP can be disadvantageous for our particular application, MAP is sensitive to the outliers. Hence, for lower adaptation ratios, where we do not work with the entire test set, if the adaptation set and the held-out set come from disjoint distributions (through random seeding), the MAP estimates can be substantially affected.
>
> * **Other baselines for test-time adaptation**:
> We would like to point out that our self-training baseline (namely, ExEnt-FT) already subsumes the entropy minimization objective proposed for test-time adaptation in Wang et al. (2021). We will make this clearer in our final version.\
> \
> Apart from this, another approach for test-time adaptation is pretraining the model on the target domain/task data using a task-independent objective (e.g., masked language modeling). However, this approach is not suitable for our setting as the target tasks are also structured (tabular in nature) and masked language modeling is not a natural task in this format. We are not aware of any other baseline strategies to compare against. \
> \
> Why do we compare against baselines built with ExEnt ? – Our evaluation focuses on showing the improvement of using TALC by forming a direct comparison to previous work. For this, ExEnt is the most natural choice since it the best current model pre-trained on the structured (tabular) training data of the CLUES dataset.
>
>
>
> * **Reproducibility Concern**: We would like to emphasize that all data as well as base models used in the paper are public. All hyperparameters for our main experiments have been provided in the Appendix as well. We will release our code on first publication, to aid reproducibility and further research in this direction.
>
> * **Typos**: Thanks for pointing out the typos. We will correct them in the final version.
> * **More details about EM algorithm**: As suggested we will provide more information about the EM algorithm (including the update equations) in the Appendix of the final version.
>
> **Reference**:
> Wang et al. 2021: Tent: Fully Test-time Adaptation by Entropy Minimization, ICLR 2021

---

### Official Review · Reviewer_U5MY · 2023-08-04

**Soundness:** 3

**Excitement:**

4: Strong: This paper deepens the understanding of some phenomenon or lowers the barriers to an existing research direction.

**Paper Topic And Main Contributions:**

This work introduces a framework to adapt language guided classifiers on new tasks. Given explanations from multiple teachers and unlabeled test examples and leveraging data programming the proposed method, TALC, showcases competitive performance when compared to prior work on real world classification tasks.

**Reasons To Accept:**

- The proposed method is novel and applicable in the real world by tackling the problem of adaptability of language-guided classifiers on unseen tasks.
- The authors compare their work against prior work, showcase strong improvement on a given dataset and analyse the pros and cons of their method.

**Reasons To Reject:**

In general the paper is well written and includes a comprehensive analysis.

**Reproducibility:**

4: Could mostly reproduce the results, but there may be some variation because of sample variance or minor variations in their interpretation of the protocol or method.

**Reviewer Confidence:**

2: Willing to defend my evaluation, but it is fairly likely that I missed some details, didn't understand some central points, or can't be sure about the novelty of the work.

---

> ### Author Rebuttal · Authors · 2023-08-28
>
> Thank you for finding our proposed method novel and applicable to real world problems. We additionally thank the reviewer for considering our paper well-written and our analysis comprehensive.

---

### Meta-Review · Area_Chair_Bjth · 2023-09-15

**Recommendation:** 3

**Metareview:**

This is an empirically sound work which proposes an approach for test-time adaptation of language-guided classifiers. In contrast to prior works, this paper uses EM to learn some parameters on unlabeled test examples to enable better aggregation. During the discussion period there were some questions as to (1) novelty compared to existing data-programming methods and (2) whether this method would extend to LLM-based models. The authors ran extensive experiments during the rebuttal period to answer (2), but questions regarding the methodological contribution above existing work remain.

---

### Decision · Program_Chairs · 2023-10-07

**Decision:**

Accept-Findings

**Comment:**

This is an empirically sound work which proposes an approach for test-time adaptation of language-guided classifiers. In contrast to prior works, this paper uses EM to learn some parameters on unlabeled test examples to enable better aggregation. During the discussion period there were some questions as to (1) novelty compared to existing data-programming methods and (2) whether this method would extend to LLM-based models. The authors ran extensive experiments during the rebuttal period to answer (2), but questions regarding the methodological contribution above existing work remain.